# Rationale and Design of the Orencia Atherosclerosis and Rheumatoid Arthritis Study (ORACLE Arthritis Study): Implications of Biologics against Rheumatoid Arthritis and the Vascular Complications, Subclinical Atherosclerosis

**DOI:** 10.3390/mps4040083

**Published:** 2021-11-15

**Authors:** Tomoaki Ishigami, Toshihiro Nanki, Takuya Sugawara, Kotaro Uchida, Hiroyuki Takeda, Tatsuya Sawasaki, Lin Chen, Hiroshi Doi, Kentaro Arakawa, Sae Saigo, Ryusuke Yoshimi, Masataka Taguri, Kazuo Kimura, Kiyoshi Hibi, Hiromichi Wakui, Kengo Azushima, Kouichi Tamura

**Affiliations:** 1Department of Medical Science and Cardio-Renal Medicine, Graduate School of Medicine, Yokohama City University, Fukuura, Kanazawa-Ku, Yokohama 236-0004, Japan; t196034g@yokohama-cu.ac.jp (T.S.); swinging_jazz_life@yahoo.co.jp (K.U.); t156052b@yokohama-cu.ac.jp (H.D.); hiroking@gamma.ocn.ne.jp (K.A.); saesaigo1019@googlemail.com (S.S.); yoshiryu@med.yokohama-cu.ac.jp (R.Y.); taguri@yokohama-cu.ac.jp (M.T.); c_kimura@yokohama-cu.ac.jp (K.K.); hibikiyo@yokohama-cu.ac.jp (K.H.); hiro1234@med.yokohama-cu.ac.jp (H.W.); azushima@yokohama-cu.ac.jp (K.A.); tamukou@yokohama-cu.ac.jp (K.T.); 2Division of Rheumatology, Department of Internal Medicine, Toho University School of Medicine, Ota-ku, Tokyo 143-8541, Japan; toshihiro.nanki@med.toho-u.ac.jp; 3Proteo-Science Center, Ehime University, Matsuyama 790-8577, Japan; takeda.hiroyuki.mk@ehime-u.ac.jp (H.T.); sawasaki@ehime-u.ac.jp (T.S.); 4Department of Cardiology, Sir Run Run Hospital, Nanjing Medical University, Long Mian Avenue 109 Jiangning, Nanjing 210011, China; mysterylin@foxmail.com

**Keywords:** atherosclerosis, rheumatoid arthritis, autoantibody, inflammation, autoimmune, clinical trial, extra-articular manifestation, abatacept, CTLA4-Ig (Cytotoxic T lymphocyte-associated antigen-4-Ig)

## Abstract

To explore the biological and immunological basis of human rheumatoid arthritis and human atherosclerosis, we planned and reported a detailed design and rationale for Orencia Atherosclerosis and Rheumatoid Arthritis Study (ORACLE Arthritis Study) using highly sensitive, high-throughput, human autoantibody measurement methods with cell-free protein synthesis technologies. Our previous study revealed that subjects with atherosclerosis had various autoantibodies in their sera, and the titers of anti-Th2 cytokine antibodies were correlated with the severity of atherosclerosis. Because rheumatoid arthritis is a representative autoimmune disease, we hypothesized that both rheumatoid arthritis and atherosclerosis are commonly developed by autoantibody-mediated autoimmune processes, leading to incessant inflammatory changes in both articular joint tissues and vessel walls. We planned a detailed examination involving carotid artery ultrasonography, measurements of adhesion molecules, such as ICAM-1 (intercellular adhesion molecule 1) and VCAM-1 (vascular cell adhesion molecule 1) for the evaluation of atherosclerosis progression, and high-throughput, high-sensitivity, autoantibody analyses using cell-free technologies, with detailed examinations of the disease activity of rheumatoid arthritis. Analyses of correlations and associations between biological markers and degrees of carotid atherosclerosis over time under consistent conditions may enable us to understand the biological and humoral immunity background of human atherosclerosis and autoimmune diseases.

## 1. Introduction

Rheumatoid arthritis (RA) is a representative autoimmune disease that affects approximately 700,000 individuals in Japan. RA causes chronic and persistent swelling, pain, and deformity of the joints of the upper and lower extremities, disturbing the activities of daily living of patients. Thus, RA is a serious and urgent issue in Japan because it affects both the individual patient’s abilities and the national workforce, and because of the national healthcare burden arising from increased social welfare expenditure (to cover the expenses of treatment, care, and so on). Thus, radical countermeasures against RA are currently required in Japan and across the world.

Atherosclerosis is a major “extra-articular manifestation” of RA, and it determines the prognosis of patients with RA. The progression of atherosclerosis precedes the manifestation of RA, possibly affecting prognosis. Currently, the availability of various anti-inflammatory drugs in addition to molecular-targeted drugs and antibody preparations has made the control of joint destruction in RA patients to a level of RA remission practically feasible. Suppression of atherosclerosis in RA patients aimed at improving the prognosis of joints and for survival is considered an important open issue to evaluate the effects of anti-RA drugs in suppressing atherosclerosis, with a focus to achieve the improvement of the anti-atherosclerotic and survival-improving effects of these drugs, in addition to the improvement of articular prognosis [1,2,3].

We established a system that enables sensitive and high-throughput measurement of autoantibodies. This system has been applied to various diseases, including atherosclerosis [4]. We have demonstrated various autoantibodies that are involved in atherosclerosis, and applications for the registration of patents based on some of the findings of our studies have been filed to patent offices across the world [5].

RA is a representative autoimmune disease. According to our studies, atherosclerosis can also be viewed as an autoimmune disease that is mediated by autoantibodies. Therefore, both diseases might have a common biological basis in that they can be characterized by persistent inflammation through autoimmune mechanisms.

Abatacept, CTLA4-Ig (Orencia, BMS, New York, NY, USA) is efficacious against RA through its inhibition of CD80/86-CD28 interactions (one of the co-stimulatory pathways between antigen-presenting cells and T lymphocytes) and inhibition of the formation of immunological synapses. Thus, abatacept is a unique drug whose efficacy is related to its activity on the more upstream action point (involved in the mechanism for the onset of autoimmune disease, characterized by persistent inflammation), unlike TNFα and IL-6 inhibitors, which act directly on inflammatory mediators. Moreover, abatacept is expected to control the common biological basis between atherosclerosis and RA, based on evidence from past studies on RA, data from our sensitive and high-throughput autoantibody assay of atherosclerosis cases, and the action mechanism of abatacept at the molecular level.

This study aimed to demonstrate that abatacept can safely and effectively reduce the activities of RA and atherosclerosis and improve the levels of biomarkers in patients with atherosclerosis accompanying RA. We also aimed to identify disease-specific autoantibody markers via our novel high-sensitivity, high-throughput autoantibody screening systems using cell-free protein synthesis technologies through the analysis of changes in autoantibody profiles, following treatment with abatacept (ORACLE Arthritis Study (Orencia Atherosclerosis Additionally, Rheumatoid Arthritis Study), UMIN000015217).

## 2. Materials and Methods

### 2.1. Target

#### Patients and Inclusion/Exclusion Criteria

Of the (1) target patients, those who met all the (2) inclusion criteria and none of the (3) exclusion criteria were considered eligible.

(1)Target patients

RA patients aged 20 years or older who met the 2010 ACR-EULAR classification criteria (Ann Rheum Dis: 2010; 69: 1580–1588) or 1987 ACR classification criteria (Arthritis Rheum 1988; 31: 315–324) and had never been treated with biological products or molecular-targeted drug therapy

(2)Inclusion criteria:
(i)Patients aged 20 years or older at the time of informed consent provision;(ii)Patients with RA that was poorly controlled with existing anti-rheumatic drugs; at the usual dosage for 3 months or more, in accordance with the criteria stipulated by the Japan College of Rheumatology. Safety criteria: white blood cell count (WBC) > 4000/mm^3^, lymphocytes > 1000/mm^3^, and testing negative for blood β-D-glucan;(iii)Patients who personally provided voluntary written consent prior to participation in the study, upon full understanding of the details of the study.


(3)Exclusion criteria:

Patients for whom abatacept was contraindicated, such as patients with a history of hypersensitivity to any of the composite ingredients of abatacept.

i.Patients with active infections or malignancies;ii.Patients who had been treated with other biological products or molecular-targeted drug therapy;iii.Pregnant women, lactating women, and patients who desired to become pregnant;iv.Patients who did not provide consent to participate in the study.

Furthermore, the following patients were excluded:i.Patients with symptomatic heart failure (NYHA class 2 or higher or EF < 40%);ii.Patients with suspected type 1 diabetes mellitus;iii.Patients scheduled to undergo coronary revascularization (patients who had undergone percutaneous transluminal coronary angioplasty, including those who underwent drug-eluting stent placement);iv.Patients who underwent coronary artery bypass surgery;v.Patients with severe liver or renal disorder;vi.Patients with a history of allergy to abatacept or drug hypersensitivity;vii.Patients with arteriosclerosis obliterans (Fontaine class III or higher);viii.Patients who were considered ineligible by the investigators for reasons other than the above reasons;

### 2.2. Study Type and Design

This was a single-arm study to compare RA activity, bilateral common carotid artery intima-media thickness (mean and max IMT), serum markers of atherosclerosis (VCAM-1, ICAM-1), and serum autoantibody profile in chronological order, from baseline to post-abatacept infusion. Autoantibody profiles during abatacept treatment was evaluated chronologically from baseline to reveal changes in RA activity, autoantibodies that show specific responses to changes in serum markers of atherosclerosis and mean/max IMT. For autoantibodies that show sensitivity to atherosclerosis, serum samples obtained from patients with atherosclerosis and healthy subjects who underwent a complete medical check-up were analyzed to validate the relationship with phenotypes, as necessary (Figure 1)

### 2.3. Outline of the Study

#### 2.3.1. Treatment Regimen Using the Study Drug

Abatacept was administered intravenously, using a standard regimen. The usual adult dosage is administered intravenously by drip. After the initial administration, the drug is administered 2 weeks and 4 weeks later, and every 4 weeks subsequently. The doses should be determined according to the table below (Table 1).

#### 2.3.2. Rules for Concomitant Medications (Therapies)

Treatment drugs that are found to be necessary by the investigators in consideration of RA treatment were co-administered.

#### 2.3.3. Rules for Dosage Reduction and Treatment Interruption

The investigators reduced the dosage and/or interrupted the treatment, according to patients’ symptoms and laboratory values.

#### 2.3.4. Patient Enrolment and Randomization Methods (Details Are Presented Below)

When enrolling patients, each investigator received informed consent forms from those who consented to participate in the study, filled out the patient enrolment form, and sent it to the Central Registration Center, Department of Medical Science and Cardiorenal Medicine, Yokohama City University Graduate School of Medicine before study initiation. This was a single-arm study without a comparator; thus, there was no randomization of enrolled patients.

#### 2.3.5. Observations and Tests/Data Collection Schedule

Data collection and analyses: patient profile at enrolment (age, gender, body height, body weight, blood pressure, pulse rate, and electrocardiography and chest radiography findings), laboratory test data (peripheral blood, blood biochemistry, urinalysis, and serology), RA activity (DASCRP28, ACR-20 response rate, ACR-50 response rate, and Simplified Disease Activity Index for RA (SDAI), Clinical Disease Activity Index (CDAI), and Boolean remission rates), bilateral mean IMT, radiographical examination of the hands and feet, assessment of serum markers of atherosclerosis, and evaluation of serum autoantibody profile. For items other than bilateral mean IMT, blood was collected and assessed at weeks 12 ± 2, 26 ± 4, and 52 ± 4. Bilateral mean IMT was calculated at weeks 26 ± 4 and 52 ± 4 W (Table 2).

For the following items, laboratory test data and electronic medical records were collected. For data collection, the investigator transcribed each assessment item in data sheets and promptly sent the transcribed data to the study secretariat by fax or e-mail. Differences in visit time points was within ±1 month. Clinical data was collected by each specialist. Blood data was processed using an automated analysis equipment by a clinical laboratory technician at each site. Radiography images of the hands and feet were taken by a radiology technician at each site, and centralized data were assessed by more than one specialist at the study secretariat. Carotid artery ultrasonography was conducted by a clinical laboratory technician at each site in accordance with the “standard method for ultrasound evaluation of carotid artery lesions” by the Japan Society of Ultrasonics in Medicine. Carotid artery ultrasonography findings in digital images were sent to the study secretariat and analyzed using IMT automatic measurement software program (Intima Scope 5.0_R_ Media Cross Co Ltd., Tokyo, Japan). To ensure objective data collection, each clinical data was collected by independent representatives at each site, maintaining patient confidentiality.

At enrolment (at baseline): age, gender, body height, body weight, BMI, smoking history, complications (hyperlipidemia, hypertension, diabetes mellitus, angina pectoris, myocardial infarction, cerebral infarction, cerebral hemorrhage, chronic renal disease, and peripheral vascular disorder), details of treatment, history of lung disease, duration of RA, history of RA treatment (histories of corticosteroid and nbDMARD treatment), stage, class, rheumatoid factor level, and anti-CCP antibody level.

General tests (at baseline, 12 ± 2 weeks, 26 ± 4 weeks, and 52 ± 4 weeks): WBC, differential WBC (neutrophils and lymphocyte ratio), albumin, ATS, alanine aminotransferase, creatinine, immunoglobulin G, total cholesterol, high-density lipoprotein cholesterol, triglyceride, blood sugar, and glycated hemoglobin levels.

RA treatment (at 52 ± 4 weeks): RA treatment for 1 year.

Assessment of RA disease activity (at baseline, 12 ± 2 weeks, 26 ± 4 weeks, and 52 ± 4 weeks): tender joints (at 28 joints), number of swollen joints (28 joints), erythrocyte sedimentation rate, C-reactive protein level, matrix metalloproteinase-3, patient global assessment (visual analog scale [VAS]), and evaluator (physician) global assessment (VAS).

Assessment of RA physical function (at baseline and 48 weeks): radiography of the hands and feet and health assessment questionnaire assessment.

Onset of serious infection (at 52 ± 4 weeks): whether serious infection occurred in a year. Details of patients who developed serious infection were taken.

Assessment of atherosclerosis (at baseline, 26 ± 4 weeks, and 52 ± 4 weeks): carotid artery ultrasonography (intima-media thickness: mean IMT, max IMT).

Treatment of complications (at 52 ± 4 weeks): treatment of complication in a year.

#### 2.3.6. Endpoints

(1)Primary endpoints

Changes in RA activity (DASCRP28, ACR-20 response rate, and ACR-50 response rate) and bilateral mean IMT

(2)Secondary endpointsChanges in the following endpoints during treatment and from baseline to post-infusion:(i)Serum markers of atherosclerosis;(ii)Atherosclerosis structural markers (FMD and ABI);(iii)Serum autoantibody profile;(iv)Blood biochemistry and serological data associated with RA;(v)Changes in radiography findings based on the Sharp score;(vi)Joint ultrasonography findings;(vii)SDAI, CDAI, and Boolean remission rates, and clinical improvement based on the health assessment questionnaire–disability index;(viii)Arterial stiffness markers (AI or PWV).


#### 2.3.7. Criteria for Study Participation Discontinuation in Individual Patients

(1)Actions to be taken at study discontinuation

When a patient was considered unfit to continue with the study, based on the reasons mentioned below, the investigator or sub-investigator (hereafter referred to as the investigators) withdrew the patient from the study. At that time, the reason for the withdrawal was explained to the patient. The investigators were required to provide treatment to the patients with good faith after withdrawal so that they will not suffer any disadvantage.

(2)Criteria for discontinuation:
(i)When a patient request to opt out of the study or withdraws consent;(ii)When serious adverse events occur, and treatment continuation is determined impossible;(iii)When the entire study is prematurely terminated;(iv)When the investigators consider it appropriate to discontinue the study for other reasons.


#### 2.3.8. Target Sample Size, Rationale for Its Determination, and Statistical Analysis Methods

(1)Target sample size (30 patients) and rationale for its determination

[Rationales for sample size determination]

Assuming that a mean difference in the DAS28 CRP score between before the intervention and 6 months after the intervention, defined as the primary endpoint, is -0.5 with a standard deviation (SD) of 1, a sample size of 25 patients is calculated to be necessary to show it with a two-sided significance level of 5% and power of 80%. By anticipating the presence of unassessable patients in the patient population, the total number of patients was set to 30 patients.

(2)Statistical analysis methods

Statistical analyses: to examine the endpoint at baseline and at weeks 4, 24, and 48, changes during abatacept treatment were evaluated using the repeated measures ANOVA method. Treatment efficacy was assessed with a significance level of *p* < 0.05.

Specifically, changes in the DAS28 CRP score, ACR-20 response rate, ACR-50 response rate, mean IMT, and biomarkers from baseline to each evaluation time point were assessed using the repeated measures ANOVA method, with a significance level of *p* < 0.05.

For high-sensitivity high-throughput screening of autoantibodies, 19 autoantibodies identified by the text-mining approach and 19 proteins, excluding overlapping proteins, selected from the analyses in other reports and the present data were defined as autoantibody markers specific to atherosclerosis and were assessed using the repeated measures ANOVA method.

Furthermore, to evaluate changes in autoantibody levels using the population approach, changes in the distribution of each antibody level were assessed as mean values with SDs using the repeated measures ANOVA method and appropriate statistical analysis methods.

#### 2.3.9. Consideration of Patients’ Rights and Method for the Protection of Personal Information

Patient involvement in the study was in compliance with the “Declaration of Helsinki (amended in October 2008)” and “Ethical Guidelines for Clinical Studies (revised on 31 July 2008; hereafter referred to as Clinical Study Ethical Guidelines).”

In handling samples related to the protocol of the study, samples were assigned numbers that were not related to the patients’ personal information to achieve patient confidentiality. In publishing study results, data enabling the identification of patients were excluded. Patients’ samples were utilized solely for the purpose of this study.

#### 2.3.10. Obtaining Informed Consent

After receiving approval from the institutional review board for the inclusion of the patient population, the investigators provided written and oral information to the patients, and the patients voluntarily provided written consent.

When obtaining information or making changes in the protocol or other relevant documents that would require the patients’ consent, the investigators promptly provided the additional information to the patients to confirm their willingness to proceed with the study, and the original written information for patients and/or other relevant documents were revised, and patient consent was reobtained.

The written information for patients contained the following information:(i)Participation in the study is voluntary; patients’ refusal to participate in the study is not associated with disadvantages, and the patient is allowed to withdraw from the study at any time;(ii)Significance (background), objectives, subjects, method, period, and planned sample size of the study;(iii)Anticipated benefits and foreseeable disadvantages of participation in the study;(iv)Handling, storage period, and method of sample disposal, including patients’ personal information, and study methods are available for inspection;(v)Handling in the cases of publication of study results and patent acquisition;(vi)Burden of study-related costs by the patients, sources of study funds, and conflict of interest;(vii)Study organization structure and consultation office (contact information) for inquiries on the study and for lodging complaints, and(viii)Actions to be taken and the availability of compensation in the event of health injuries to patients.

## 3. Results

The protocol of the ORACLE Arthritis Study was approved by the institutional research board of Yokohama City University Hospital (B141001011) on 1 November 2014 and the outline of the protocol was registered in UMIN National Database on 1 October 2014 (UMIN000015217). Following this, we started the enrolment of participants at individual hospitals and out-patient clinics, according to the protocol. On 23 February 2015, 27 May 2015, and 22 February 2016, 25%, 50%, and 100% of the registration, respectively, had been performed. The last visit of the subjects was scheduled for 5 April 2017. Forty-one participants (men 7, women 34; average age 67.8 ± 11.9) were enrolled; eventually, 35 participants completed the protocol. The remaining six patients quit the study for the following reasons: patient’s request (*n* = 2), insufficient treatment effect determined by the researcher, complication of pneumonia, intention to become pregnant (*n* = 1 each), and transfer to another hospital (*n* = 1).

## 4. Discussion

To date, atherosclerotic diseases are mainly treated by two strategies: (i) typical critical medical treatment, which comprises urban emergency transport and optimal end-vascular therapy in a hospital and is for emergency catastrophic conditions caused by ruptured foamy plaque in the arterial wall, and (ii) risk factor modification by optimal medical treatment of diabetes mellitus, hyperlipidemia, hypertension, and CKD. Both strategies are complementarily applied by catheter intervention operators and regular medical physicians in developed countries. Various healthcare technologies and factors, such as medical electronics, biochemical laboratory testing, nursing system, and essential medical personnel, commonly support these systems explicitly or implicitly. In the past, atherosclerotic diseases were considered to originate from vascular injury of various sources (“response to injury” hypothesis). Currently, scientific progression has revealed that inflammatory processes are pivotal in the formation of atherosclerotic lesions throughout the vascular wall, with a myriad of cellular components and oxidized lipid burden. Therefore, atherosclerotic diseases are currently regarded as “inflammatory” diseases. Subsequently, anti-inflammatory agents have become therapeutic agents for atherosclerosis, based on the abovementioned conceptual change. The CANTOS study on canakinumab, which is a human anti-IL-1β monoclonal antibody developed by Novartis, is a representative clinical trial involving anti-inflammatory agents for the treatment of human atherosclerotic diseases [6].

Atherosclerosis is a major “extra-articular manifestation” of RA and determines the prognosis of RA patients [7,8]. Subjects with RA manifested a higher cardiovascular risk than the cardiovascular risk predicted by the Framingham risk score [9]. From the Canadian RA registry, the high mortality among RA patients under 45 years old was due to both respiratory and circulatory diseases [8]. When coronary plaque characteristics detected in autopsy were compared between subjects with RA and controls without RA, the subjects with RA showed more vulnerable plaques in the left anterior descending artery and more frequent medial inflammation of the left circumflex artery [10]. Accordingly, ischemic heart disease in subjects with RA commonly involved sudden cardiac death, and mortality after myocardial infarction was higher in the subjects with RA than in those without RA [7]. Therefore, treatment of subjects with RA should be focused on joint survival as well as mortality and morbidity due to cardiovascular diseases.

Subjects with RA showed more frequent CV complication than was predicted. The prevalence of diabetes mellitus in subjects with RA was high and might have been associated with CV complications [11], but there is a paradoxical inverse association between CV complications, lipid, and BMI [12,13]. Individuals with high BMI have lower mortality than thinner subjects with RA [12], and lower total cholesterol and LDL cholesterol levels are associated with increased cardiovascular risk [13]. Pathologic mechanistic insight into this paradox is important, but we need to consider non-traditional CV risk which can be managed in subjects with RA. By cohort observations, both CDAI (Clinical Disease Activity Index) [14] and anti-citrullinated peptide/protein antibodies in RA were significantly correlated with CV events [15]. These findings and other considerations prompted us to perform the ORACLE Arthritis Study to examine the detailed biological dissections between subclinical atherosclerosis in subjects with RA and the immunological disease activity in these subjects. In past experiments using animal models, studies that utilized ApoE-/-mice treated with homocystetine, or a model of femoral artery cuff injury revealed antiatherosclerotic effects of abatacept [16,17]. Therefore, abatacept is expected to suppress atherosclerosis in RA patients and improve prognosis. To date, however, few related studies have been conducted in humans [18,19]. Existing antibody preparations and biologics have been applied to only limited cases (patients with orphan disease, autoimmune disease, malignant disease, and so on). The present study has the potential of not only expanding the indications of abatacept to common diseases, but can also serve as a factor triggering the attempt of controlling the biological base for common diseases (atherosclerosis, CKD, and so on) with a population approach to the autoantibody profile.

## 5. Conclusions

We designed, planned, and completed the single-arm, observational ORACLE Arthritis study to clarify the immunological and biological bases of human RA and human atherosclerosis using high-sensitivity, high-throughput autoantibodies screening technologies.

## Figures and Tables

**Figure 1 mps-04-00083-f001:**
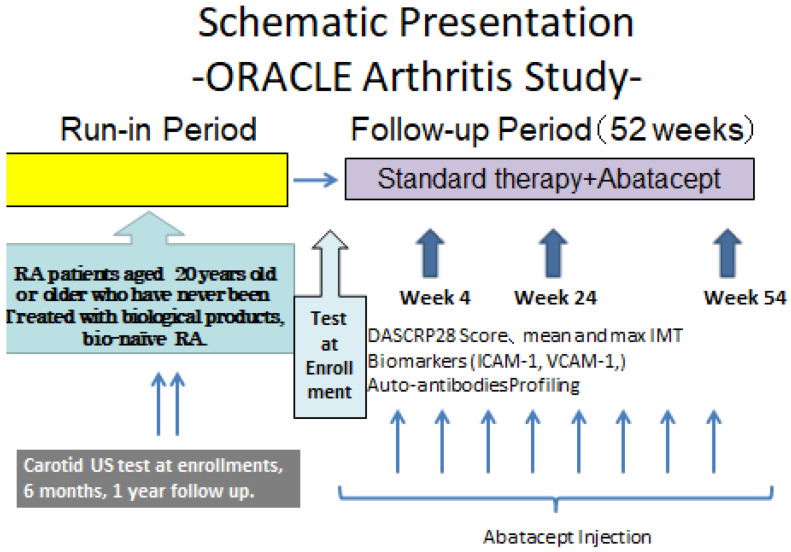
Schematic Presentation of ORACLE Arthritis Study.

**Table 1 mps-04-00083-t001:** Treatment regimen using the study drug.

Body Weight of Patient	Dose	Number of Vials
Less than 60 kg	500 mg	2
60 to 100 kg	750 mg	3
More than 100 kg	1 g	4

**Table 2 mps-04-00083-t002:** Data Collection Schedule.

		Follow-Up Period (52 ± 4 Weeks)				
Time point	Baseline	0 W	4 W	12 ± 2 W	26 ± 4 W	52 ± 4 W
Informed consent	◯					
Patient background characteristics/RA activity	◯		◯	◯	◯	◯
Blood pressure and pulse rate	◯	◯	◯	◯	◯	◯
Blood biochemistry	◯	◯	◯	◯		◯
Mean IMT		◯			◯	◯
Radiography examination (hands and feet)/joint ultrasonography findings	◯			◯	◯	◯
Biomarker autoantibody	◯			◯	◯	◯
Adverse events		← ◯ →				

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
