# Peer review of "Rationale and Design of the Orencia Atherosclerosis and Rheumatoid Arthritis Study (ORACLE Arthritis Study): Implications of Biologics against Rheumatoid Arthritis and the Vascular Complications, Subclinical Atherosclerosis"

_mps, 2021, doi:10.3390/mps4040083_

Round 1
Reviewer 1 Report
Very intersting paper and an important contribution to the field.
My suggestion and doubts are about blinding, as DAS28 and IMT are not exactly objectives and can vary according to the examinator
Evaluation of DAS28 --> Will the examinator be blinded to the results of the IMT ? I think that is very important and must be specifficaly described in the methodology
Evaluation of IMT --> Will the US examinator be blindend to the DAS28 results ? Again, I think that that is very important and must be specifficaly described in the methodology.
Will be the same examinator of the IMT in the baseline and prospective evaluation ? If possible, the same evaluator blinded will be adequately. Ideally, two observators at the evaluations (in a way to evaluate divergences) as IMT observations could vary among observators
Author Response
Response to Reviewers’ Comments
Reviewer #1
Comment: Very interesting paper and an important contribution to the field. My suggestion and doubts are about blinding, as DAS28 and IMT are not exactly objectives and can vary according to the examinator. Evaluation of DAS28 –> Will the examinator be blinded to the results of the IMT ? I think that is very important and must be specifically described in the methodology. Evaluation of IMT –> Will the US examinator be blinded to the DAS28 results ? Again, I think that that is very important and must be specifically described in the methodology. Will be the same examinator of the IMT in the baseline and prospective evaluation ? If possible, the same evaluator blinded will be adequately. Ideally, two observators at the evaluations (in a way to evaluate divergences) as IMT observations could vary among observators
Response: Thank you for your comment on our manuscript. We really appreciate your comment on our manuscript. As the reviewer commented, we believe our current experiments would show an additional evidence on a current rheumatology and cardiology in view of commonly underlying immunological background. As commented by the reviewer, we completely agree that objective data collection about DAS28 and IMT are necessary to be clearly warranted. Therefore, we carefully designed ORACLE Arthritis Study about objective data collection. Evaluations of DAS28 and IMT during the study were performed by responsible attending physicians and responsible ultrasonographers independently with blinded to each data. Additionally, as described in the manuscript p 5 as followings;“Carotid artery ultrasonography findings in digital images were sent to the study secretariat and analyzed using IMT automatic measurement software program (IntimaScope 5.0R Media Cross Co Ltd, Tokyo, Japan).”, the results of IMT measurements are evaluated by central-lab with least intra-observer variance. We added following sentence in the end of the paragraph in this revised manuscript.(highligeted in red)
“For the following items, laboratory test data and electronic medical records were col-lected. For data collection, the investigator transcribed each assessment item in data sheets and promptly sent the transcribed data to the study secretariat by fax or e-mail. Differences in visit time points was within 1 month. Clinical data was collected by each specialist. Blood data was processed using an automated analysis equipment by a clinical laboratory technician at each site. Radiography images of the hands and feet were taken by a radiol-ogy technician at each site, and centralized data were assessed by more than one specialist at the study secretariat. Carotid artery ultrasonography was conducted by a clinical labor-atory technician at each site in accordance with the “standard method for ultrasound evaluation of carotid artery lesions” by the Japan Society of Ultrasonics in Medicine. Ca-rotid artery ultrasonography findings in digital images were sent to the study secretariat and analyzed using IMT automatic measurement software program (IntimaScope 5.0R Media Cross Co Ltd, Tokyo, Japan). To ensure objective data collection, each clinical data is collected by independent representatives at each site with concealment.” (page 5, line 173~)
Reviewer 2 Report
I suggest to insert arterial stiffness (augmentation index and pulse wave velocity) as vascular markers .
Furthermore i suggest to insert if possible only patients with monotherapy to evaluete only the abtacept effects on atherosclerosis.
To improve the study i will compare abtacept to metotrexate or adalimumab or tocilizumab
Please provide a standard dose of steroide during the follow up
Author Response
Response to Reviewers’ Comments
Reviewer #2
Comment: I suggest to insert arterial stiffness (augmentation index and pulse wave velocity) as vascular markers. Furthermore i suggest to insert if possible only patients with monotherapy to evaluate only the abtacept effects on atherosclerosis. To improve the study i will compare abtacept to metotrexate or adalimumab or tocilizumab. Please provide a standard dose of steroide during the follow up
Response: Thank you for your comment on our manuscript. We really appreciate your comment on our manuscript and your proposal to improve our study. We added “(viii) Arterial stiffness markers (AI or PWV)” in the revised manuscript. (p.6, line 223) It is possible to evaluate only the abatacept effects on atherosclerosis as one of post-hoc analyses for the study. For the second recommendation, it is not allowed to use different biologics other than abatacept for RA treatment in this study. Dose of steroid are not determined by the protocol, but information about medical treatment of RA including dose of steroid are collected during the study.
This manuscript is a resubmission of an earlier submission. The following is a list of the peer review reports and author responses from that submission.